# The Role of Vitamin D in Autoimmune Thyroid Diseases: A Narrative Review [note 1]

**DOI:** 10.3390/jcm12041452

**Published:** 2023-02-11

**Authors:** Agata Czarnywojtek, Ewa Florek, Krzysztof Pietrończyk, Nadia Sawicka-Gutaj, Marek Ruchała, Ohad Ronen, Iain J. Nixon, Ashok R. Shaha, Juan Pablo Rodrigo, Ralph Patrick Tufano, Mark Zafereo, Gregory William Randolph, Alfio Ferlito

**Affiliations:** 1Department of Pharmacology, Poznan University of Medical Sciences, 60-806 Poznan, Poland; 2Chair and Department of Endocrinology, Metabolism and Internal Medicine, Poznan University of Medical Sciences, 60-355 Poznan, Poland; 3Laboratory of Environmental Research, Department of Toxicology, Poznan University of Medical Sciences, Dojazd 30 Street, 60-631 Poznan, Poland; 4Voivodal Specialistic Hospital in Olsztyn, 10-561 Olsztyn, Poland; 5Department of Otolaryngology-Head and Neck Surgery, Galilee Medical Center, Azrieli Faculty of Medicine, Bar-Ilan University, Tel Awiw 5290002, Israel; 6Department of Otorhinolaryngology Head and Neck Surgery, Edinburgh Royal Infirmary, Edinburgh EH8 9YL, UK; 7Head and Neck Service, Memorial Sloan-Kettering Cancer Center, New York, NY 10065, USA; 8Department of Otolaryngology, Hospital Universitario Central de Asturias, Fundación de Investigación e Innovación Biosanitaria del Principado de Asturias, and Instituto Universitario de Oncología del Principado de Asturias, University of Oviedo, 33011 Oviedo, Spain; 9Multidisciplinary Thyroid and Parathyroid Center, Head and Neck Endocrine Surgery, Sarasota Memorial Health Care System, Sarasota, FL 34239, USA; 10Department of Head & Neck Surgery, MD Anderson Cancer Center, Houston, TX 77005, USA; 11Department of Otolaryngology Head and Neck Surgery, Harvard Medical School, Boston, MA 02114, USA; 12Coordinator of the International Head and Neck Scientific Group, 35100 Padua, Italy

**Keywords:** vitamin D, Hashimoto thyroiditis, Graves’ disease, Riedel thyroiditis, postpartum thyroiditis

## Abstract

Vitamin D (VitD) deficiency has garnered significant attention in contemporary medical research. Although the canonical biological activity of VitD manifests itself mainly in the regulation of calcium-phosphorus metabolism, recent studies show that, thanks to the presence of numerous receptors, VitD may also play an important role in regulating the immune system. VitD deficiency has been demonstrated to impact autoimmune disease, coeliac disease, infections (including respiratory/COVID-19), and patients with cancer. Recent studies also show that VitD plays a significant role in autoimmune thyroid diseases (AITDs). Many studies have shown a correlation between low VitD levels and chronic autoimmune thyroiditis – Hashimoto thyroiditis (HT), Graves’ disease (GD), and postpartum thyroiditis (PPT). This review article, therefore, describes the current state of knowledge on the role of VitD in AITDs, including HT, GD, and PTT.

## 1. Introduction

Vitamin D (VitD) deficiency in people around the world has become an important health problem. VitD significantly affects the regulation of calcium-phosphorus metabolism, and currently also plays an extraordinary role in the immune system. Moreover, many studies have shown that VitD is also a prohormone that affects immune-related diseases and disorders [1], including psoriasis, type 1 diabetes [2], multiple sclerosis, rheumatoid arthritis, systemic sclerosis, systemic lupus erythematosus, Sjögren’s syndrome, mixed connective tissue disease, antiphospholipid syndrome, coeliac disease [3,4,5,6], Mycobacterium tuberculosis [7,8,9,10], sepsis and critical illness [11], infections, respiratory infections, COVID-19 [12], metabolic diseases including primary biliary cirrhosis [3,4] and inflammatory bowel diseases [13,14,15]. Low levels of serum VitD are correlated with an increased risk of developing other diseases, such as cardiovascular diseases, neurocognitive dysfunction [16], and, very importantly, neoplasms [17], including prostate cancer, colon cancer, breast cancer, leukaemia [8] and thyroid cancer [18].

This article aims to describe the latest studies on the correlation between VitD levels and autoimmune thyroid diseases (AITDs), based on a retrospective narrative review.

The current state of knowledge outlined in this review article is based on a query of literature available in PubMed, Sinomed, and China National Knowledge Infrastructure, conducted from October 1980 to December 2022. The following terms used in combination with ‘vitamin D’ were searched for: ‘thyroid disease’, ‘Hashimoto’s thyroiditis’, ‘Graves’ disease’, ‘postpartum thyroiditis’, ‘hypothyroidism’ and ‘hyperthyroidism’.

## 2. Vitamin D: Prohormone, Metabolism and Function

VitD is a steroid compound, which consists of two secosteroids differing in the structure of the side chains: ergocalciferol (VitD2) and cholecalciferol (VitD3) [19,20]. While VitD2 occurs in plants and fungi (synthesised from ergosterol), VitD3 can be found in animals [21]. VitD in the human body is traditionally classified as a typical vitamin, but it is also believed to play the role of a prohormone because it takes a biologically active form of 1α,25-dihydroxycholecalciferol after metabolic transformations [22].

In the human body, VitD3 is generated in the epidermis through a photochemical transformation of 7-dehydrocholesterol after absorption of ultraviolet radiation (wavelength 280–315 nm) energy [21,23,24]. This process produces previtamin-D, which is transformed into VitD and two biologically inactive compounds, lumisterol and tachysterol, as a result of thermal conversion. It is important because VitD has no side effects following long-term exposure to solar radiation [19]. It should be noted that VitD2 is present in fungi and plants (synthesised from ergosterol) [21], while VitD3 is contained in fish oil and eggs in addition to the skin [21,24].

VitD is hydroxylated in the liver to produce 25(OH)D (25-hydroxycholecalciferol) – calcidiol in the first step. The catalysts for this reaction are cytochrome P450 monooxygenases (CYPs), which include: CYP27A1, CYP3A4 and CYP2R1 [25]. It is this 25(OH)D that is a biochemical measure of the body’s supply of VitD [21,25].

On the other hand, in the kidneys, hydroxylation occurs in the 1α and/or 24 positions. In the case of 1α-hydroxylation, 1α,25-dihydroxyvitamin D (1α25OH2D), also known as calcitriol, is formed. The reaction of this hydroxylation is carried out by the enzyme 25(OH)D-1α-hydroxylase (CYP27B1) [26]. In the case of 24-hydroxylation by 25(OH)D-24-hydroxylase (CYP24A1), 25(OH)D and 1α25OH2D are inactivated, preventing the accumulation of toxic levels of VitD [21,24,26].

It should be noted that the activity of CYP24A1 plays one of the main roles in the action of VitD [27]. The activity of 1α-hydroxylase (CYP27B1) and/or 24-hydroxylase (CYP24) is increased by parathyroid hormone (PTH), PTH-related peptide (PTHrP), prostaglandins, hypocalcaemia and hypophosphataemia, and inhibited by fibroblast growth factor-23 (FGF23) and by 1,25(OH)2D alone. CYP24A1 is the only identified 24-hydroxylase that is regulated in the opposite way to renal 1α-hydroxylase as it is induced by 1α25OH2D and FGF23 [21].

The role of VitD, and specifically 1,25(OH)2D, is to increase the absorption of calcium from the intestines. PTH ensures the proper serum level of calcium. During VitD deficiency, PTH levels increase [28]. The activity of VitD is based on processes both dependent [9] and independent of gene transcription [29]. The effective action of VitD primarily involves binding to the nuclear vitamin D receptor (VDR). This receptor binds to the retinoid receptor X (dimerization process). The resulting heterodimer binds to VitD-responsive genes [9,30]. On the other hand, the rapid action of VitD involves regulating the intracellular calcium level, but independently of gene transcription [29,30].

Płudowski et al. [16] have demonstrated a completely new role of VitD. Namely, in addition to influencing the osteoarticular system, VitD is a prohormone that affects metabolic diseases, the cardiovascular system, infections, autoimmune diseases, neurocognitive disorders and, very importantly, the neoplastic process.

## 3. Vitamin D in AITDs

The process of autoimmunity involves the formation of autoantigens in the body – in this case, to the thyroid gland. Under normal conditions, when the human body shows tolerance, the process of autoimmunity does not take place. However, when this tolerance is broken, autoimmunity occurs, resulting in the production of autoantigens such as thyroid peroxidase (TPO), thyroglobulin (Tg) and thyroid-stimulating hormone (TSH) receptor (TSH-R). As a result, in the process of autoimmunity to these autoantigens, three antibodies are produced, two associated with Hashimoto’s thyroiditis (HT): anti-TPO and anti-Tg, and one with Graves’ disease (GD): anti-TSH-R (TRAbs). Anti-TPO and anti-Tg can be associated with GD, but the most important marker for GD is anti-TSHRs (TRAbs) [31].

In both diseases, there is a lymphocytic infiltration in the thyroid gland (thyroid parenchyma). However, while the follicular cells remain intact in GD, in the case of HT the thyroid parenchyma is destroyed, which leads to hypothyroidism [32]. Additionally, there is increased ‘expression’ (recruitment) of Th1 lymphocytes and stimulation of interferon-γ and tumour necrosis factor-α (TNF-α), which stimulate secretion of the cytokine CXCL10 by thyroid cells. This results in positive feedback and initiation of the autoimmune process, which is then consolidated [31].

The recruitment of Th1 lymphocytes is accompanied by stimulation of B lymphocytes, which are located in secondary lymphatic vesicles in the tissue of the thyroid gland. As a result, autoantibodies are spontaneously produced in AITDs [32].

Coperchini et al. [33] have shown that VitD increases the expression of ACE-2 on both the mRNA and protein levels in thyroid cells. This increased ACE-2 expression takes place owing to VitD in connection with IFN-γ. Thus, patients with AITD may potentially have increased expression of ACE-2 (IFN-γ) or decreased expression of ACE-2 (VitD deficiency). According to the hypothesis of these authors, patients with AITD may have a more severe course of COVID-19, which is associated with lower ACE-2 expression.

The development of AITDs is influenced by endogenous factors (genetic factors, oestrogens, adipokines and body mass index) [31,34] and exogenous factors (nicotinism, iodine, selenium and VitD deficiency, infections, stress, and nutrition) [31].

It should be emphasised that VitD plays an essential role in AITDs as it modulates the immune system, enhancing the innate immune response [35], and acts as an immunomodulator in autoimmune diseases, including GD and HT [36].

## 4. A Correlation between TSH, anti-TPO, anti-Tg and VitD Levels in Healthy People and Those with AITDs

In healthy adults, in the euthyroid phase, a strong inverse correlation has been shown between TSH and VitD levels (a metabolite of VitD–calcidiol (25(OH)D_3_), with TSH levels being the highest in the autumn-winter period and the VitD levels being the highest in the spring and summer period [37].

Mackawy et al. [38] have also confirmed this correlation (between VitD and TSH), additionally showing a high incidence of hypovitaminosis and hypocalcaemia in patients with hypothyroidism. Chailurkit et al. [39] have confirmed the above data in young people as well as in middle-aged and older men, where negative anti-thyroid antibody titres have been additionally demonstrated [40]. A further study in Korea showed that iodine excess was associated with thyroid dysfunction only in people with VitD deficiency [41]. Mazokopakis et al. [42] have shown an inverse correlation between serum 25(OH)-vitamin D levels and anti-TPO antibody production in 218 patients with hyperthyroidism and normal thyroid function. Patients with VitD deficiency had significantly higher anti-TPO antibody levels than those without this deficiency. A significant decrease (20.3%) in serum anti-TPO levels was observed in 186 patients who received oral VitD supplementation (1200–4000 IU/day) for four months.

Chaudhary et al. [43] studied 100 patients who had recently been diagnosed with AITD and discovered that those in the lowest VitD quartile had the highest levels of anti-TPO antibodies (*p* = 0.084). A three-month follow-up study showed significant decreases in the levels in patients who received VitD supplementation for eight weeks (60,000 IU/week). Comparing this group to the patients who were not administered VitD supplementation, there was a 46.73% decrease versus 16.6%, and this difference was statistically relevant (*p* = 0.028). The former also reported a higher number of responders (≥25% decrease in anti-TPO levels) than the latter (68% vs. 44%; *p* = 0.015).

### Vitamin D Levels in AITDs

Numerous scientific studies have shown a correlation between low VitD levels and an increased risk of AITDs. It has been shown that if the VitD levels are ≤10 ng/mL (≤25 nmol/L), AITDs often occur, most often in the form of HT and an increase in anti-thyroid antibodies [44]. In GD, lower levels of VitD were found compared to HT, and an inverse correlation between 25(OH)D_3_ levels and anti-thyroid antibody titres [45]. These studies were additionally confirmed in 2015 by Wang et al. [31], who studied 1782 patients with AITDs and a control group (n = 1821). This correlation has also been shown in an independent, separate comparative study of patients with GD and HT [46].

## 5. Analysis of Individual AITDs: Hashimoto’s Thyroiditis, Graves’ Disease, and Postpartum Thyroiditis

### 5.1. Hashimoto’s Thyroiditis (HT)

Many recent studies have shown a correlation between VitD levels and HT, and these results have been compared to those of healthy people [47,48,49,50].

In 2011, low VitD levels were found for the first time in patients with hypothyroidism due to HT compared to patients with euthyreosis (HT) [45]. Later studies, conducted in other centres on other populations, further confirmed the correlation between lower VitD levels and the risk of HT [36,48,49,50,51,52,53,54]. Interestingly, this relationship has been additionally proven based on VitD levels, TSH levels, anti-TPO titre levels, anti-Tg and the patient’s age.

A study by Muscogiuri et al. [55] on people over 65 years of age has shown that when VitD levels are <20 ng/mL (~50 nmol/L), AITDs and anti-TPO titres are more frequent. Similar results concerning VitD deficiency have been obtained in children with HT [56,57,58,59].

A randomised clinical trial study conducted by Chahordoli et al. [60] included 42 women with HD, who were divided into VitD and placebo groups. The first group received 50,000 IU of VitD, whereas the second one was given placebo pearls each week for three months. Laboratory parameters such as VitD, calcium, anti-TPO, anti-Tg, thyroid hormones, and TSH were assessed at the beginning and end of the study using enzyme-linked immunosorbent assays. Significant reductions in anti-Tg and TSH were observed in the first group after treatment, but there were no significant differences in anti-TPO levels between the two groups (*p* = 0.08). Similarly, thyroid hormone levels changed insignificantly. It was thus concluded that VitD supplementation was useful in decreasing disease activity. Nevertheless, additional evidence confirmed by well-controlled longitudinal trials is crucial to determine its relevance in clinical practice.

A meta-analysis that included 25 studies comprising 2695 cases and 2263 controls showed that HD patients had lower VitD serum levels compared to the control groups; however, there was significant heterogeneity between the studies (Cohen’s D = 0.62; 95% CI 0.89–0.34; *p* = 1.5 × 10^−5^). Compared to the control groups, in individuals with VitD-deficiency, an odds ratio of 3.21 of having HD (1.94–5.3; *p* = 5.7 × 10^−6^) was determined. These results were consistent in all the considered studies: European and Asian, adult and paediatric, and moderate- and high-quality studies. Higher differences in 25(OH)D values between the groups were related to near-equatorial latitudes (<35° N/S, *p* = 3.4 × 10^−4^) and moderate-income economies (gross national income 1000 < USD < 12,000, *p* = 0.012). In univariate meta-regression, lower Cohen’s d correlated with higher latitude (*p* = 0.0047) or higher mean body mass index (*p* = 0.006 in ten studies). Gross national income (*p* = 3.5 × 10^−6^) and mean serum thyrotropin in the affected patients (*p* = 0.017 in 21 studies) revealed nonlinear moderation. The main finding indicates that there is a significant correlation between HD and VitD, and partly explains previous mixed evidence, showing what factors are involved in heterogeneity and under what conditions the relationship is the strongest [61].

According to Chao et al. [62], patients with HT have reduced VitD levels, and TSH is an independent risk factor for HT. TSH is negatively correlated with VitD levels. FT3 and FT4 levels are positively correlated with VitD levels.

Bozkurt et al. [49] have found a correlation between the severity of VitD deficiency, the duration of HT, the volume of the thyroid gland and antibody titres. These studies have also been confirmed by other scientists: Giovinazzo et al. [48], Arslana et al. [63], Shin et al. [64] and Goswami et al. [65]. However, Wang et al. [66] made a very interesting observation, showing a negative correlation between VitD25 and anti-Tg levels but no correlation in the case of anti-TPO. Some studies have shown no correlation between VitD and TSH, anti-TPO and anti-Tg levels in children [57,59] and adults [41,53,56,65]. However, a recent study by Cvek et al. [67] indicates that there is no correlation between VitD and HT. Nonetheless, VitD levels may subtly go down due to overt hypothyroidism.

### 5.2. The role of Vitamin D in Graves’ Disease (GD)

Many historic publications have shown no relationship between VitD and GD. On the other hand, numerous recent studies have indicated a correlation between low VitD levels and GD (Table 1) [45,68,69,70,71,72,73,74].

Additionally, Zhang et al. [40] were the first to show the relationship between VitD levels and TRAb titres. Already in 2015, a significant link between VitD deficiency and GD had been demonstrated [31,77,78].

Yasuda et al. [70] showed that VitD deficiency very significantly prevailed (*p* < 0.05) in GD (65.4%) compared to the control group (32.4%). A significant correlation was found between VitD levels, calcium levels (r = 0.49; *p* < 0.05), and unchanged parathyroid hormone concentrations (r = −0.50, *p* < 0.05). Furthermore, significant correlations between thyroid volume and 25(OH)–VitD levels were observed (r = −0.45; *p* < 0.05) but the same could not be determined for TRAb values and thyroid function.

Interestingly, Ahn et al. [73] have shown that GD relapses frequently in such situations. VitD levels may be important in response to treatment, with lower levels associated with a lower likelihood of remission [71] and a higher relapse rate [79] when antithyroid medications are used.

In their randomised prospective study, which included 60 adults with GD (aged 20–40), Sheriba et al. divided them into two groups. In the first one, 20 patients received 30 mg of methimazole daily, and in the second one, 40 patients were administered the same dose of methimazole and 200,000 IU of VitD3 monthly for three months. A follow-up study was conducted for three months. All participants had a form of hypovitaminosis: 73.9% of males and 54.1% of females with deficiency and 26.1% of males and 45.9% of females with insufficiency. A strong correlation was found between VitD levels, thyroid volume, and the degree of exophthalmos. Lowering thyroid volume was observed upon supplementation in the second group and positive effects on the exophthalmos degree were also found [80].

Behera et al. studied patients with hyperthyroidism, 23 of whom received doses of 60,000 IU of VitD per week for eight weeks and then the same dose once per month for four months. TPO antibody and thyroid hormone serum levels were determined again after six months. VitD levels increased significantly from 15.33 ± 5.71 ng/mL to 41.22 ± 12.24 ng/mL. TPO antibody titres significantly went up from 746.8 ± 332.2 to 954.1 ± 459.8 IU/mL (*p* = 0.006), and TSH values significantly decreased from 7.23 ± 3.16 to 3.04 ± 2.62 mIU/L (*p* = 0.01) [81].

Meta-regression and sensitivity analysis were also performed by combining the effect sizes from 26 studies by Xu et al. [77]. A pooled connection of standard mean difference (SMD) = −0.77 (95% CI: −1.12, −0.42; *p* < 0.001) was found, and random effect analysis favoured the low VitD levels. Meta-regression indicated that heterogeneity was mainly affected by the assay technique (*p* = 0.048). GD patients were more predisposed to VitD deficiency compared to the control group (OR = 2.24, 95% CI 1.31–3.81). This finding showed high heterogeneity. Thus, low VitD levels were proven to increase GD risk [77].

Interestingly, radioactive iodine therapy (RIT) was not successful when serum VitD levels were <20 ng/mL [78]. These results have been confirmed by Ahn et al. [73], who have demonstrated low effectiveness of RIT in cases of VitD deficiency (<20 ng/mL). Moreover, Komarovskiy K et al. [79] have shown that hypocalcaemia develops after RIT, but also occurs after treatment with thyreostats and strumectomy [74].

It should be noted that Jyotsna et al. [75] Kim et al. [52] and Ke et al. [53] have not found any relationship between the concentration of VitD and GD.

Planck et al. [76] analysed recent-onset GD in 292 patients and 2305 controls by determining their VitD levels. They examined correlations between GD/Graves’ ophthalmopathy and VDR single nucleotide polymorphisms, and between vitamin D-binding protein and CYP27B1, respectively, in 708 patients and 1,178 controls. The results indicated significantly lower VitD values (*p* < 0.001) in the GD group (55.0 ± 23.2 nmol/L) compared to healthy patients (87.2 ± 27.6 nmol/L). Values of thyroid hormones such as free thyroxine or free triiodothyronine, thyrotropin receptor antibodies, relapse upon antithyroid medication discontinuation, and Graves’ ophthalmopathy showed no correlation with VitD levels during diagnosis.

In our opinion, however, too few patients were examined in these studies. In the future, following the example of Planck et al. [76], a larger research group should be studied to increase the reliability of research results.

A recent meta-analysis by Taheriniva et al. [82] has shown no relationship between the level of VitD and GD, except for the group of patients ≥40 years of age. Despite abundant evidence from previous years denying the correlation between the concentration of VitD and GD, this relationship is currently increasingly demonstrated.

Taking into account the above considerations and in line with what Rotondi et al. [83] have concluded, to formulate firm conclusions, it is necessary to conduct further studies specifically designed at assessing the role of VitD deficiency and the effect of its correction in patients with Graves’ disease.

### 5.3. Analysis of Women with Postpartum Thyroiditis (PPT)

Postpartum thyroiditis (PPT) is characterised by autoimmune destructive thyroiditis that occurs in the first year postpartum with an incidence of 5% [84]. Analyses of women with PPT by Ma et al. [71] and Krysiak et al. [85] have also shown a correlation between lower VitD levels and AITD induction.

Krysiak et al. [85] showed that in women with post-partum thyroiditis, thyroid peroxidase and thyroglobulin antibody titres correlated with PTH levels despite adjusting for VitD. Interestingly, they also found out that l-thyroxine treatment increased VitD levels and reduced PTH levels exclusively in hypothyroid women with post-partum hypothyroidism. This is in contrast with the improved thyroid function observed in women with both autoimmune and non-autoimmune hypothyroidism. Moreover, the higher serum VitD and lower PTH levels associated with l-thyroxine treatment correlated with reduced thyroid antibody titres, but not with changes in TSH and thyroid hormones. These results suggest that although normalisation of TSH and thyroid hormone levels is undoubtedly the most significant effect of l-thyroxine treatment, VitD homeostasis seems to depend more on its anti-inflammatory effects.

Nguyen and Mestman [84] have proved that women who have undergone PPT are at risk of permanent hypothyroidism.

### 5.4. De Quervain Thyroiditis and VitD

Calapkulu et al. [86] examined 170 patients with de Quervain thyroiditis and 86 healthy people, and proved that the concentration of VitD was significantly lower in patients with de Quervain thyroiditis than in healthy people. These authors have shown no relationship between the level of VitD and the prognosis of the disease. In addition, they have proved that VitD deficiency may be associated with decreased immunity and the occurrence of infectious diseases of the upper respiratory tract, which may be the cause of de Quervain thyroiditis.

### 5.5. The role of vitD in Patients with Riedel’s Thyroiditis

There are few publications directly describing the effect of VitD levels on Riedel’s thyroiditis. However, the subject of VitD often appears in the context of the treatment of Riedel’s disease, due to the complication of hypoparathyroidism. VitD together with calcium eliminates the negative effects of parathyroid hormone deficiency [87,88,89,90].

## 6. Vitamin D Supplementation in Thyroid Diseases

Research shows that VitD supplementation is essential in preventing AITDs. Research by Wang et al. [31] has shown that the standard administration of VitD is a minimum of 2.000 IU/day to 60.000 IU/week. However, the latest studies have shown that VitD levels ≥50 ng/mL (125 nmol/L) reduce the risk of hypothyroidism by up to 30% [41].

Therefore, the administration of high doses of VitD in patients with hypothyroidism rapidly improved thyroid function [89,91,92]. It has been proven that VitD supplementation can also prevent the development of thyroid diseases. Moreover, in their meta-analysis and systematic review, Wang et al. [31] have proved that VitD supplementation significantly reduces anti-TPO and anti-Tg levels. These results have been confirmed by Koehler et al. [92] and this was especially true of the level of anti-TPO antibodies.

The Endocrine Society’s guidelines for VitD supplementation indicate that VitD levels >30 ng/dL is optimal in people at risk of VitD deficiency. Additionally, it has been proven that VitD levels up to 100 ng/mL (250 nmol/L) are not toxic but very safe and do not cause hypercalcaemia [93]. In Poland, the guidelines for VitD supplementation are based on those specified by the Endocrine Society and define its optimal levels between 30–50 ng/mL (75–125 nmol/L) and the maximum safe levels up to 100 ng/mL (250 nmol/L) [40,93].

## 7. Discussion

VitD is a steroid hormone traditionally associated with the metabolism of phosphocalcium. The discovery of pleiotropic expression of its receptor, and the enzymes involved in its metabolism, led to the exploration of other roles of this vitamin. The effect of VitD on AITDs has been extensively studied. Most existing data confirm that VitD deficiency is correlated with a greater tendency to develop HT, GB and PPT and/or higher titres of antibodies associated with these diseases [36,40,49,50,51,53,71,72,76,78,82]. Unfortunately, there are too few observations for rare diseases such as de Quervain’s disease or Riedel’s thyroiditis.

However, some discrepancies do not confirm this relationship completely. Even if a correlation is assumed between VitD, or VitD polymorphisms, and AITDs, it is still unclear whether this reflects a pathological mechanism, a causal relationship, or a consequence of the autoimmune process, which further complicates the conclusions [38,39,40,51,52,53,54,55,78]. While there is some inconsistency in the research results obtained so far, most of the data confirm that VitD deficiency is correlated with AITDs. Therefore, there are several publications where no relationship has been found between VitD deficiency and AITDs. Kim et al. [52] and Ke et al. [53] and have proven low levels of VitD in Hashimoto’s disease but shown no association with GD.

Wang et al. [31] have made a very interesting observation in their systematic review and meta-analysis, suggesting that VitD supplementation appears to significantly lower anti-TPO levels without any side effects. More recently, Koehler et al. [92] have retrospectively analysed 933 patients with autoimmune thyroiditis and found greater reductions in anti-TPO antibodies in the subgroup of 58 patients whose initially inadequate VitD levels (<30 ng/mL) increased compared to the control group who maintained VitD levels below the threshold.

Ensuring the correct VitD levels may also be important for a proper response to the treatment of thyroid diseases, as its lower levels are associated with a lower likelihood of remission [46] and a higher rate of relapse [31].

Therefore, research is needed to fully understand the meaning of VitD supplementation in AITDs [23]. It should be emphasised that elevated PTH values, and slightly decreased calcium levels, can be associated with VitD deficiency in the absence of parathyroid disease.

## 8. Conclusions

Most scientific studies have found that VitD deficiency is common in people with AITDs, suggesting it is likely to play some role in their development. Numerous studies, including multicentre, randomised clinical trials, recommend VitD supplementation in AITD patients. Moreover, VitD has also been shown to have a beneficial effect on the effectiveness of therapy when used in conjunction with many medications used in both hypothyroidism and hyperthyroidism.

## Figures and Tables

**Table 1 jcm-12-01452-t001:** Relationship between Graves’ disease and level of vitamin D.

Study	Year	Case Group (F/M)	Control Group (F/M)	Age of Case Group	Age of Control Group	Level of Vitamin D Case	Level of Vitamin D Control	Level of TRAb (IU/l)	Association
Bouillon et al. [72]	1980	77/27	-	39	-	28 ± 11 (ng/L)	42 ± 13 (ng/L)	-	+
Yasuda et al. [70]	2012	26/0	46/0	37.3 ± 13.0	44.3 ± 18.1	14.4 ± 4.9 (ng/mL)	17.1 ± 4.1 (ng/mL)	19.9 ± 18.1	+
Jyotsna et al. [75]	2012	62/18	62/18	36.33 ± 11.15	36.42 ± 10.40	12.67 ± 6.24 (ng/mL)	10.99 ± 7.05 (ng/mL)	-	-
Unal et al. [45]	2014	27 (F&M)	124 (F&M)	44.6	44.6	14.9 (4–39) (ng/mL)	19.9 (9–122.7) (ng/mL)	-	+
Ma et al. [71]	2015	70 (F&M)	70 (F&M)	40.04 ± 15.24	41.99 ± 13.31	31.71 ± 13.10 (nmol/L)	41.33 ± 14.48 (nmol/L)	14.16 (5.81–27.92)	+
Kim et al. [52]	2016	148 (F&M)	407 (F&M)	-	48.6 ± 13.3	39.3 ± 22.0 (ng/mL)	39.9 ± 21.5 (ng/mL)	-	-
Ke et al. [53]	2017	30/21	31/20	39.79 ± 1.73	36.48 ± 1.68	81.77 ± 5.60 (nmol/L)	83.49 ± 6.24 (nmol/L)	-	-
Planck et al. [76]	2018	246/46	921/1384	45.5 ± 13.1	59.4 ± 7.2	55.0 ± 23.2 (nmol/L)	87.2 ± 27.6 (nmol/L)	-	+
Mangaraj et al. [69]	2019	54/30	25/17	35.25 ± 9.70	32.41 ± 9.71	19.22 ± 8.95 (ng/mL)	23.81 ± 12.46 (ng/mL)	19.45 ± 12.12	+

## Data Availability

Data sharing not applicable.

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
