# Peer review of "The Role of Vitamin D in Autoimmune Thyroid Diseases: A Narrative Review†"

_jcm, 2023, doi:10.3390/jcm12041452_

Round 1

Reviewer 1 Report

Dear authors,

I compliment you on this well written review concerning the role of vitamin D in autoimmune thyroid diseases. While the article is very strong and well organized, there is excessive use of causal language without the evidence to show any conclusive evidence by the end of the paper. Here are some minor suggestions:

1) Consider rewording parts of the abstract, as midway the statement "Recent studies also show that VitD plays a significant role in autoimmune thyroid diseases (AITDs)" clearly contradicts with the later statement "While there is no clear evidence that VitD plays a significant role in AITDs"

2) In the section titled "Correlation between TSH and vitamin D levels in healthy people and AITDs" I believe the subtitle "The levels of TSH and VitD in healthy people" is misplaced.

3) Should a similar Table 1 be created for all your subgroup analyses?

4) Are these following two statements not somewhat contradictory to one another? "Interestingly, radioactive iodine therapy (RIT) was successful when serum VitD levels were < 20 ng/ml [69]. These results have been confirmed by Ahn et al. [68], who have demonstrated low effectiveness of RIT in cases of VitD deficiency (< 20 ng/ml)."

5) In response to the results of the meta-analysis by Taheriniva et al, the following question "Is it a coincidence that we have a VitD deficiency in developed countries?" appears to be out of place without appropriate elaboration and evidence presented by the authors. If the meta-analysis was performed recently, and there were no major flaws in their analysis, wouldn't this include the recent studies related to "this relationship is currently increasingly demonstrated."? Does Taheriniva's meta-analysis comment on this?

6) In many subgroup analyses, the authors claim that other authors have "proved" the relationship between vitD deficiency and the AITD in question. Then later state "there is no clear evidence that VitD plays a significant role in AITDs". but also state "Research shows that VitD supplementation is essential in preventing AITDs." I would be careful with the use of such conflicting language leaving the author confused with the directionality of the paper.

7) In the section concerning vit D doses, the authors claim "However, the latest studies have shown that VitD levels ≥ 50 ng/ml (125 nmol/l) reduce the risk of hypothyroidism by up to 30%." but only go on to site one single study

Author Response

Manuscript ID: jcm-2158403

Title: The role of vitamin D in autoimmune thyroid diseases: a narrative review

We would like to thank the Referees’ for their careful review of our manuscript and for specific advice and comments to improve the quality of our work. We have carried out a revision of the manuscript and we believe the paper has been significantly improved.

According to the Reviewers’ suggestion, the manuscript has been carefully checked and corrected. The changes in the manuscript have been marked in green colour. Below we address all changes by the points raised by the Reviewers.

Reviewer 1

Firstly, we would like to express our deepest thanks to the Reviewer for devoting time to reviewing our manuscript, the corrections and suggestions. We have carried out a revision of the manuscript and we believe the paper has been improved.

The Reviewer’s comment: Consider rewording parts of the abstract, as midway the statement "Recent studies also show that VitD plays a significant role in autoimmune thyroid diseases (AITDs)" clearly contradicts with the later statement "While there is no clear evidence that VitD plays a significant role in AITDs".

The Authors’ answer: Thank you for such specific comments, they helped us to improve our manuscript. Sentence: "While there is no clear evidence that VitD plays a significant role in AITDs", which was included in the summary has been deleted. As Reviewer 1 has noticed this sentence was completely contradictory and there was no logic in it.

The Reviewer’s comment: In the section titled "Correlation between TSH and vitamin D levels in healthy people and AITDs" I believe the subtitle "The levels of TSH and VitD in healthy people" is misplaced.

The Authors’ answer: Dear Reviewer, thank you very much for your comment. Subtitle, as suggested by Reviewer 1 "The levels of TSH and VitD in healthy people" is out of place. This sentence has been deleted.

The Reviewer’s comment: Should a similar Table 1 be created for all your subgroup analyses?

The Authors’ answer: Dear Reviewer, thank you for your comment. Surely, it will be valuable to create a table for Riedel's thyroiditis or De Quervain Thyroiditis and VitD, but there are only a few publications, while in the case of Hashimoto's disease there are a lot of different tables in many articles, but we deliberately did want to do this, so that we would not be accused of plagiarism.

The Reviewer’s comment: Are these following two statements not somewhat contradictory to one another? "Interestingly, radioactive iodine therapy (RIT) was successful when serum VitD levels were < 20 ng/ml [69]. These results have been confirmed by Ahn et al. [68], who have demonstrated low effectiveness of RIT in cases of VitD deficiency (< 20 ng/ml)."

The Authors’ answer: Thank you for such specific comments, they helped us to improve our manuscript. In this case, a linguistic error occurred during the translation. Of course, it was in both cases low effectiveness of RIT in cases of VitD deficiency (< 20 ng/ml).

The Reviewer’s comment: In response to the results of the meta-analysis by Taheriniva et al, the following question "Is it a coincidence that we have a VitD deficiency in developed countries?" appears to be out of place without appropriate elaboration and evidence presented by the authors. If the meta-analysis was performed recently, and there were no major flaws in their analysis, wouldn't this include the recent studies related to "this relationship is currently increasingly demonstrated."? Does Taheriniva's meta-analysis comment on this?

The Authors’ answer: Indeed, Reviewer 1 is right that this sentence is out of place "appears to be out of place". In fact, observations should have taken more time to claim that. Therefore, this statement was deleted, and we wrote a new sentence, citing the comments of the Reviewer 2.

The Reviewer’s comment: In many subgroup analyses, the authors claim that other authors have "proved" the relationship between vitD deficiency and the AITD in question. Then later state "there is no clear evidence that VitD plays a significant role in AITDs". but also state "Research shows that VitD supplementation is essential in preventing AITDs." I would be careful with the use of such conflicting language leaving the author confused with the directionality of the paper.

The Authors’ answer: Thank you for such specific comments, they helped us to improve our manuscript. we show this discrepancy on purpose to show that sometimes "scholar’s opinions are divided". That's why we explained this "such conflicting language" because these studies were done in the past and were not so detailed. Therefore, the sentence "there is no clear evidence that VitD plays a significant role in AITDs" has been deleted.

The Reviewer’s comment: In the section concerning vit D doses, the authors claim "However, the latest studies have shown that VitD levels ≥ 50 ng/ml (125 nmol/l) reduce the risk of hypothyroidism by up to 30%." but only go on to site one single study.

The Authors’ answer: Dear Reviewer, thank you very much for your comment. There were many such studies, but we wanted to present the most recent studies (recommendations). Therefore, we did not change anything in this fragment, but added other publications in the further part of the publication (green color, e.g. Mazokopakis et al., GaluÈ™ca, D et al.).

Reviewer 2 Report

In the present review article, Czarnywojtek et al. Focused on the consequences of VitD deficiency in terms of regulation of the immune system. In particular this review encompasses several studies that analized the role of Vit D in Autoimmune thyroiditis (AITDs), including HT, GD and PTT. The Authors final message is that no clear evidences support that VitD plays a significant role in AITDs, however VitD deficiency significantly affect patients with even mild thyroid diseases.

This Review is well written and the topic is of potential interest for not only endocrinologists but also for all clinicians. However, some recent litterature data are missing.

Firstly, i would suggest to the Authors to re-analyze the literature to include more studies on VitD role in HT, GD and PTT. Following, you can find some suggestions to improve the manuscript.

-          The Authors described Vitamin D by writing three short pargraph. In order to make this part more fluent for the readers, I would suggest to match the three paragraphs titled Vitamin D as a prohormone, The metabolism of vitamin D, The function of vitamin D“  into one chapter

-           The Authors said that vitamin D also play a role in COVID-19. In this regard, a recent original study demonstrated that Vitamin D up-regulates the ACE-2 (the receptor of SARS COV-2) mRNA and protein levels on the cell membrane of thyroid cells. I would suggest to also include this effect of Vitamin D on ACE-2, when the Authors mentioned the role of Vitamin D in COVID-19. (doi: 10.1007/s40618-022-01857-9).

-          The paragraph  “Correlation between TSH and vitamin D levels in healthy people and AITDs” seems to lack of some crucial details that are included in the quoted studies. For reason, I would suggest to expand this paragraph by discussing the results of these articles in a more appropriate manner.

-          In the paragraphs regarding  Hashimoto’s thyroiditis (HT) and   The role of vitamin D in Graves’ disease (GD) more details should be given about the role of vitamin D in these pathologies. Given the importance  of these topic, these sections appears to be too short. For this reason, I would suggest provide more details discussed in these recent studies. I would recommend to include these articles: doi: 10.3390/medicina58020194,  doi: 10.3389/fendo.2020.00004, doi: 10.3390/nu13082793.

-           In the chapter about Graves’s Disease the Author concluded with the following sentence: “Is it a coincidence that we have a VitD deficiency in developed countries?” . I would suggest to improve this part by discussing this article titled : “Vitamin D deficiency in patients with Graves' disease: probably something more than a casual association”.doi: 10.1007/s12020-012-9776-y.

-          In the post partum chapter i would suggest to also include this article doi: 10.1038/ejcn.2016.56. Epub 2016 Apr 6.

Author Response

Manuscript ID: jcm-2158403

Title: The role of vitamin D in autoimmune thyroid diseases: a narrative review

We would like to thank the Referees’ for their careful review of our manuscript and for specific advice and comments to improve the quality of our work. We have carried out a revision of the manuscript and we believe the paper has been significantly improved.

According to the Reviewers’ suggestion, the manuscript has been carefully checked and corrected. The changes in the manuscript have been marked in green colour. Below we address all changes by the points raised by the Reviewers.

Reviewer 2

Firstly, we would like to express our deepest thanks to the Reviewer for devoting time to reviewing our manuscript, the corrections and suggestions. We have carried out a revision of the manuscript and we believe the paper has been improved.

The Authors’ answers:

  1. Firstly, the publication was supplemented by more studies about the role of vitamin D in HT, GD and PTT, as suggested by the Reviewer to improve the manuscript.

  1. Secondary, the three paragraphs titled “Vitamin D as a prohormone”, “The metabolism of vitamin D” and “The function of vitamin D” were merged into one chapter.

  1. Thirdly, as suggested by Reviewer 2, effects of Vitamin D on ACE-2 have been included and attached by referenced publication (Coperchini et al.).

  1. Fourth, the “Correlation between TSH and vitamin D levels in healthy people and AITDs” has been enriched with the extension of this paragraph.

  1. Fifth, the paragraphs about the role of VitD in Hashimoto's thyroiditis (HT) and Graves' disease (GD) were enhanced with more details, including publications (Mazokopakis et al., Chaudhary et al., Yasuda et al., Chao et al., Cvek et al.).

  1. In the chapter about Graves’s Disease, as it was suggested by Reviewer 1 sentence: “Is it a coincidence that we have a VitD deficiency in developed countries?” has been delated. In response to Reviewer 2’s suggestion, this part of our review was improved by overview of the article: „Vitamin D deficiency in patients with Graves’ disease: probably something more than a casual association”.

We would like to express our deepest thanks to the Reviewers for all efforts and contribution to the improvement of our manuscript.
